# Synthesis and Structure-Chirality Relationship Analysis of Steroidal Quinoxalines to Design and Develop New Chiral Drugs

**Rashid Mehmood** [1,*], **Naghmana Rashid** [2] , **Shakir Ullah** [2], **Maria John Newton Amaldoss** [3] and **Charles Christopher Sorrell** [1]

[1] School of Materials Science and Engineering, UNSW Sydney, Sydney, NSW 2033, Australia; c.sorrell@unsw.edu.au

[2] Department of Chemistry, Allama Iqbal Open University, Islamabad 44000, Pakistan; naghmana.rashid@aiou.edu.pk (N.R.); shakirullah75@gmail.com (S.U.)

[3] Adult Cancer Program, Lowy Cancer Research Centre, Prince of Wales Clinical School, UNSW Sydney, Sydney, NSW 2033, Australia; m.amaldoss@unsw.edu.au

\* Correspondence: r.mehmood@unsw.edu.au

**Abstract:** Of the utmost importance of chirality in organic compounds and drugs, the present work reports structure-chirality relationship of three steroidal quinoxalines, which were synthesised by condensing diaminobenzenes with cholestenone. All the compounds were purified and characterised by varying analytical tools prior to their chiroptical analysis by circular dichroism (CD) technique. The substituent groups on quinoxalines contributed to determining the chiroptical properties of the compounds. The positive Cotton effects have been observed in the CD spectra of unsubstituted and methyl-substituted quinoxalines, which indicated their P helicity. Importantly, chloro-substituent on quinoxalines produced different CD behaviour, which can be attributed to the presence of three lone pairs of electrons on Cl atom. The present work provides guidelines for determining the chiral properties of steroidal quinoxalines, which can be useful to design and develop potential molecules of biological importance.

**Keywords:** chirality; circular dichroism; helicity; steroids; quinoxalines

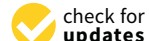



## 1. Introduction

The 2016 Nobel Prize in Chemistry, awarded to Prof. Bernard L. Feringa and colleagues, highlights the modern technological applications of chirality, which has been ignored in recent years [1]. Chirality, molecular geometry, is an intrinsic property of molecules and ions which determines their effectiveness and specificity [2,3]. The extreme case of this was observed in 1957, when a racemic mixture of thalidomide, morning sickness alleviator, reportedly affected thousands of newborn babies worldwide. It was prescribed for pregnant women during the first trimester, but it prevented the growth and nourishment of the fetus, resulting in severe birth defects [4]. Later, it was identified that the R-enantiomer of thalidomide is sedative while S-enantiomer has teratogenic effects [5]. Importantly, since human physiological systems are selective in interacting with racemic drug, each enantiomer of the racemate metabolises via distinct mechanism to induce different biological effects. Thus, one enantiomer of a racemic mixture may initiate beneficial actions, while the other may be harmful for the body [6].

More than half of the approved chemical drugs are chiral, and a wide range of bio-active compounds under-development also are chiral in nature [7]. Off these compounds, organic heterocycles are the potent agents possessing anticancer properties. Imidazole, benzimidazole, thiazole, triazole, oxazole, quinoxalines, indole, pyrimidine etc. are the important moieties which have been used with other bio-active molecules for cancer drug-development [8–15]. Naturally occurring heterocycles also have been isolated from plant and animal to possess

anticancer properties. A natural bis-steroidal heterocycle, cephalostatin, isolated from marine tube worm Cephalodiscus gilchristi, have shown potential growth inhibition of leukemia, neuroblastoma, lung, and colon cancer cells [16,17]. Among these natural and synthetic organic heterocycles, a benzene-pyrazine system, quinoxaline, has generated a significant interest owing to its wide spectrum of biological properties [18–24]. Particularly, anticancer effects of quinoxalines have been explored against leukemia, melanoma, glioblastoma, ovarian cancer, renal cancer, lung cancer, breast cancer etc [25–29]. Importantly, several quinoxalines containing compounds, including echinomycin, levomycin, thequinox, caroverine or spasmium, olaquindox, sulfaquinoxaline, topoisomerase IIB inhibitor (XK469) chloroquinoxaline sulphonamide, and 6-cyano-7-nitroquinoxaline-2,3-dione are, currently used as approved drugs [30]. They are either the quinoxalines-derivatives or combine product of quinoxaline and other biologically active molecules such as peptides [30–32].

Consequently, considering the potential medical application of quinoxalines, present work reports the synthesis and chiral analysis of steroidal quinoxalines. Purification and structural characterization of materials were done by using varying analytical tools and chiral properties were analysed by using circular dichroism (CD) spectropolarimetry. The structure-chirality relationship of these compounds provides the understanding to design and develop chiral steroidal quinoxaline-based molecules.

## 2. Materials and Methods

### 2.1. Materials and Instruments

Cholesterol (purity 99%), potassium sodium tartrate (purity 99%), anhydrous magnesium sulphate, 1,2-diaminobenzene (purity 99.5%), aluminium isopropoxide (purity 98.0%), sodium carbonate (purity 99.0%), 4-methyl-1, 2-diaminobenzene (purity 98.0%), 4-chloro-1, 2-diaminobenzene (purity 97%), sodium hydride (60% in mineral oil), petroleum ether, ethyl acetate methanol, chloroform, cyclohexanone, and toluene were purchased from Sigma-Aldrich. Characterisation of the synthesised compound was done by using varying analytical instruments including melting point apparatus (MP; Gallenkamp, Sanyo MPD350, West Midlands, UK), nuclear magnetic resonance spectrometer (NMR; Bruker Avance 300–400 MHz, Billerica, MA, USA), Fourier transform infra-red spectrophotometer (FTIR; Perkin-Elmer, Waltham, MA, USA), mass spectrometer (MS; Varian MAT 312, Palo Alto, CA, USA and Bruker micrOTOF-Q$^{TM}$, Billerica, MA, USA), and circular dichroism spectropolarimeter (CD; JASCO J-810, Tokyo, Japan).

### 2.2. Synthesis and Characterisation

Cholest-3-en-4-one (I): Anhydrous toluene (500 mL) was introduced in 1 L double-neck round bottomed flask, which was sealed with reflux assembly setup under inert atmosphere. 40 g of cholesterol was added to flask followed by the addition of 200 mL of cyclohexanone. After this, in a separate beaker, a solution of aluminium isopropoxide (12 g) in anhydrous toluene (80 mL) was prepared and added dropwise to the flask by using dropping-funnel which was sealed with reflux assembly. Initially while magnetic stirring, the reaction mixture became cloudy and yellow colour, which was heated at 80 °C for 8 h. The progress of the reaction was monitored at different intervals. Reaction assembly was dismantled, and the mixture was cooled at room temperature for 1 h. After cooling, about 160 mL of saturated solution of potassium sodium tartrate was added to the reaction mixture and the organic layer became clear and orange, then the mixture was steam distilled until 240 mL of toluene was distilled. The residue mixture was extracted with chloroform three times. Extract was washed with water, dried with anhydrous magnesium sulphate, and the solvent was removed by using rotary evaporator. The yellow oily residue was dissolved in 60 mL of anhydrous methanol and placed in an ice-salt bath for 24 h. The final white solid product was filtered, dried in a vacuum desiccator and stored in closed vessel for further use. Yield (76%), MP (80 °C), Rf value (0.8; petroleum ether: ethyl acetate, 8:1), FTIR (1680 cm$^{-1}$; keto group), 2930 cm$^{-1}$ (aliphatic methylene groups), MS (384 $m/z$; M$^{+}$), $^{1}$H-NMR [CDCl$_3$; 5.737 ppm (s, 1H), 2.43~2.29 ppm (m, 2H), 2.01~1.91 ppm (m, 2H),

1.82 ppm (m, 1H), 1.59~1.31 ppm (m, 23H), 1.26 ppm (m, 6H), 1.11 ppm (m, 3H)], $^{13}$C-NMR [CDCl$_3$; 199.8 ppm (carbonyl carbon), 171.9 ppm (α-carbon), 123.7 ppm (β-carbon)].

5α-cholest-3-eno-[3,4-b]-quinoxaline (II): Anhydrous toluene (50 mL) was introduced in 250 mL double-neck round bottomed flask, which was sealed with reflux assembly setup under inert atmosphere. Equimolar (1.2 mmole) amount of cholest-3-en-4-one **(I)** and 1, 2-diaminobenzene was added to flask. The mixture was magnetically stirred and refluxed for 1 h. After this, 20 mL of toluene was distilled out and reaction was allowed to stir at 80 °C for 24 h. The progress of the reaction was monitored at different intervals. Reaction assembly was dismantled, and the mixture was cooled at room temperature for 1 h. After cooling, toluene was removed by using rotary evaporator. The dark brown residue was dissolved in 10 mL of anhydrous methanol and purified by column chromatography using petroleum ether: ethyl acetate (5:1) solvent system. The final product was dried in a vacuum desiccator and stored in closed vessel for further use. Yield (57%), MP (165 °C), Rf value (0.45; petroleum ether: ethyl acetate, 5:1), FTIR (1538 cm$^{-1}$ quinoxaline moiety C=C), MS (472 *m/z*, M$^+$), $^1$H-NMR [CDCl$_3$; 7.68~7.92 ppm (m, 4H, C-6', C-7', C-8', C-9'), 2.33 ppm (m, 3H, C-5(α –H) and C-2) 2.00~0.70 ppm (m, cholestane nucleus peaks)], $^{13}$C-NMR [CDCl$_3$; 130.86 ppm (C-3 and C-4), 128.8 ppm (C-5', C-10'), 126.61 ppm (C-6', C-7', C-8', C-9')].

5α-cholest-3-eno-[3,4-b]-7'(8´)-methylquinoxaline (III): Anhydrous toluene (50 mL) was introduced in 250 mL double-neck round bottomed flask, which was sealed with reflux assembly setup under inert atmosphere. After this, 1.2 mmole of 4-methyl-1, 2-diamino benzene and 1.2 mmole cholest-3-en-4-one (I) was added to flask and reaction was allowed to stir at 80 °C for 24 h. The progress of the reaction was monitored at different intervals. Reaction assembly was dismantled, and the mixture was cooled at room temperature for 1 h. After cooling, toluene was removed by using rotary evaporator. The dark brown residue was dissolved in 10 mL of anhydrous methanol and purified by column chromatography using petroleum ether: ethyl acetate (5:3) solvent system and reparative thin layer chromatography. The final product was dried in a vacuum desiccator and stored in closed vessel for further use. Yield (35%), MP (195 °C), Rf value (0.39; petroleum ether: ethyl acetate, 5:3), FTIR (1535 cm$^{-1}$ quinoxaline moiety C=C), MS (486 *m/z*, M$^+$), $^1$H-NMR [CDCl$_3$; 7.53–7.51 ppm (d, 2H, C-8' and C-9' when CH$_3$ at C-7'), 7.06-7.04 ppm (d, 2H, C-6' and C-7' when CH$_3$ at C-8'), (d, 2H, C-7', C-8', C-9'), 6.554 ppm (s, 1H, C-6'or C-9') 2.48~2.46 ppm (s, 3H, C-11' when at C-7' or C-8'), 2.40 ppm (m, 3H, C-5(α –H) and C-2)], $^{13}$C-NMR [CDCl$_3$; 135.4 and 135.6 ppm (C-3 and C-4), 131.7 and 134.9 ppm (C-5', C-10'), 109.9–108.5 ppm (C-6', C-9'), 123.5-123.6 ppm (C-7', C-8'), 22.54, and 22.81 ppm (C-11' when at C-7' or C-8')].

5α-cholest-3-eno-[3, 4-b]-7´(8´)-chloroquinoxaline (IV): The synthesis procedure for (IV) is identical to the preceding (II) and (III) except the use of different starting material 4-chloro-1, 2-diamino benzene and use of sodium hydride to deprotonate amine of the starting materials prior to condensation with (I). Yield (23%), Rf value (0.5; petroleum ether: ethyl acetate, 5:3), FTIR [1540 cm$^{-1}$ (quinoxaline moiety C=C), $^1$H-NMR [CDCl$_3$; 7.68-7.66 ppm (d, 2H, C-8' and C-9' when Cl at C-7'), 7.42–7.40 ppm (d, 2H, C-6' and C-7' when Cl at C-8'), 5.9 ppm (s, 1H, C-6' or C-9'), 4.02 ppm (m, 3H, C-5(α –H) and C-2)], $^{13}$C-NMR [CDCl$_3$; 123.63 and 123.46 ppm (C-3 and C-4), 119.1 and 118.6 ppm (C-7', C-10'), 109.9–108.5 ppm (C-6', C-9'), 112.4-112.3 ppm (C-7', C-8'), 2.54, and 22.81 ppm (C-11' when at C-7' or C-8')].

CD analysis: Samples (II), (III), and (IV) were prepared in acetonitrile at a concentration of $2.05 \times 10^{-4}$ mol/L, $1.91 \times 10^{-4}$ mol/L, and $3.28 \times 10^{-4}$ mol/L, respectively. 1-mm and 0.5-mm rectangular cells were used to measure near UV-CD analysis at wavelength range of 200–400 nm.

## 3. Results and Discussion

In the present work, Oppenauer oxidation reaction strategy was used to synthesize, previously reported [33,34], cholest-4-en-3-one (I), which was then utilized to generate different steroidal quinoxalines, as shown in Figure 1.

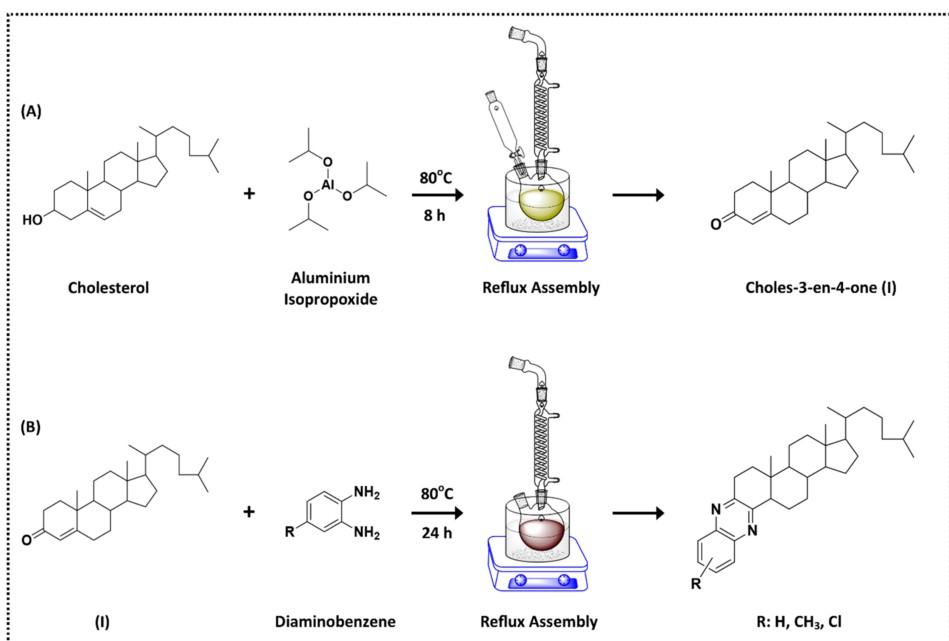

**Figure 1.** Synthesis of steroidal quinoxalines: (**A**) conversion of cholesterol to cholest-4-en-3-one (I) by Oppenauer oxidation, (**B**) condensation of diaminobenzene, 4-methyl-1,2-diaminobenzene, and 4-chloro-1,2-diaminobenzene with cholest-4-en-3-one (I).

The condensation of unsubstituted diaminobenzene and cholestenone was straight forward reaction, which was conducted under inert atmospheric reflux conditions using anhydrous toluene. Initially, the same strategy was used to condense substituted diaminobenzene but possibly, the substituent group on diaminobenzene hindered the condensation mechanism. Consequently, the use of strong base, sodium hydride facilitated the condensation, resulting in the product formation. The detailed synthesis procedures for these reactions are mentioned in Supplementary Information. Mechanistically, sodium hydride removed one of the protons from amino group of diaminobenzene molecules, which accelerated the attack of amine at carbonyl carbon of cholestenone. The possible mechanisms of these reactions are presented in Figure 2.

The purification of the products was done by column, thin layer, and preparative thin layer chromatography techniques, where petroleum ether and ethyl acetate solvent systems were used for separation. The compounds were characterised by Fourier transform infrared spectroscopy (FTIR), mass spectrometry (MS), and nuclear magnetic resonance spectroscopy (NMR), the spectra of which are shown in Supplementary Information Figures S1–S15. The data for the structure elucidation of the synthesised compounds are explained in preceding Experimental section. Figure 3 represents the structures and positioning of the carbon atoms of these synthesised compounds.

The chiral properties of (II), (III), and (IV) were analysed by CD technique, the data of which are presented in the following Table 1 and Figure 4 (individual CD spectra are shown in the Supplementary Information Figures S16–S18).

**Table 1.** CD data for steroidal quinoxalines (II), (III), and (IV).

| Compound | Solvent | Conc. [mol/L] | Cell [mm] | CD Cotton Effects [mdeg] | | |
| --- | --- | --- | --- | --- | --- | --- |
| | | | | 1st Band (Wavelength) | 2nd Band (Wavelength) | 3rd Band (Wavelength) |
| (II) | Acetonitrile | $2.05 \times 10^{-4}$ | 1 | 9.7 (238 nm) | −1.3 (282 nm) | - |
| (III) | Acetonitrile | $1.91 \times 10^{-4}$ | 1 | 13.0 (240 nm) | 2.4 (287 nm) | - |
| (IV) | Acetonitrile | $3.28 \times 10^{-4}$ | 1 | 3.9 (212 nm) | −1.9 (244 nm) | −0.016 (359 nm) |
| (IV) | Acetonitrile | $1.75 \times 10^{-4}$ | 0.5 | 3.6 (212 nm) | −1.4 (244 nm) | −0.013 (359 nm) |

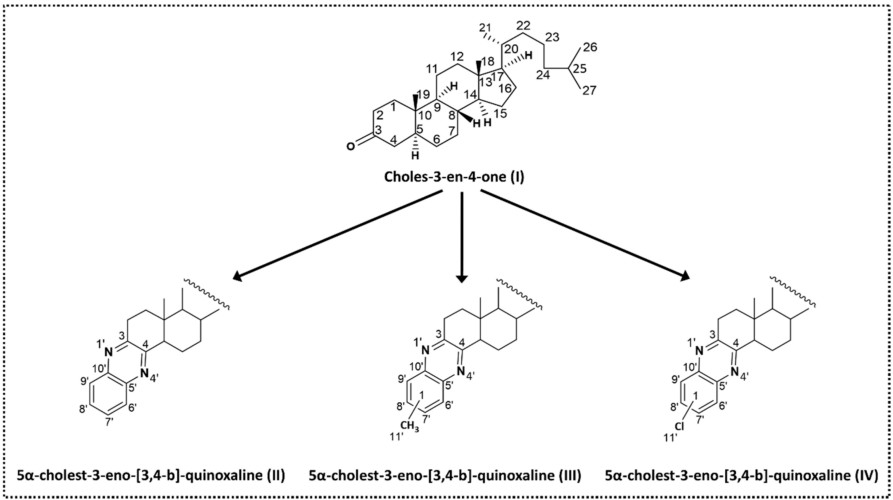

**Figure 2.** Reaction mechanism for the synthesis of steroidal quinoxalines: (i) attack of amino group at carbonyl carbon, (ii) proton transfer, (iii) loss of water molecule, (iv) attack of amine group at C=C, (v) transfer of proton from N to C, (vi) proton rearrangement, (vii) oxidation to loose protons from N; (viii) activation of amine group of substituted diaminobenzene using NaH.

**Figure 3.** Structural representation of synthesised compounds along with the carbon and nitrogen atom positioning.

To determine the helicity, Snatzke and co-workers established that steroidal heterocycles can be divided into three spheres [33–35]. According to them, if first and second sphere are achiral then sector rule or helicity rule can be applied. In terms of chromophore systems, the quinoxaline moiety of (**II**), (**III**), and (**IV**) belongs to the benzene chromophores with a chiral second sphere, according to Snatzke's terminology. For this type of benzene chromophore, a simple helicity rule can be applied [35–37]. Therefore, in case of 5α-cholest-3-eno-(3,4-b)-quinoxaline (**II**), the chromophore quinoxaline forms the first sphere. The non-aromatic ring A condensed with the first sphere comprises second sphere. The methyl

group at C-10 and the second non-aromatic ring comprise third sphere, as shown in Figure 5a. Owing to pi–pi* and *n*-pi* transitions [33], quinoxalines are expected to generate absorptions at different wavelengths with positive and negative Cotton effects [36]. Also, for quinoxalines type of system, it has been established that depending on substituents and solvent for analysis, two to three CD bands can be expected [36]. In case of (**II**) and (**III**), the major 1st CD band is observed at 238 and 240 nm, respectively, and both bands have shown the positive Cotton effect. Importantly, if pseudoaxial substituents are not present at the benzylic carbon atoms [38], positive Cotton effect within the $^1B_{2u}$ transition (α-band) leads to *P-helicity* of the non-aromatic ring while negative Cotton effect within same transition leads to *P-helicity* [38–40]. Looking through quinoxaline chromophore, it can be seen in Figure 5b that ring A is in half chair conformation, whereas rings B and C maintain their chair conformation, resulting in molecular stability. As there are no pseudoaxial substituents at the benzylic carbon atoms and major CD bands for (**II**) and (**III**) show positive Cotton effect values, the P-helicity can be assigned to both (**II**) and (**III**).

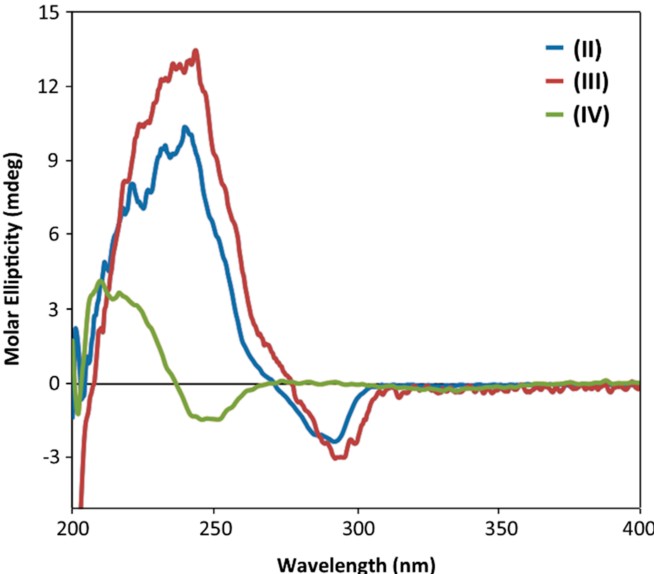

**Figure 4.** Comparison of CD analysis of compound (**II**), (**III**), and (**IV**).

For 2nd CD band in (**II**) and (**III**), the peaks observed at 282 and 287 nm, respectively, and both bands have shown the negative Cotton effect. While it is evident that the helicity of the chiral ring attached to the chromophore is not affected by the negative Cotton effect of 2nd band, because the helicity of the chiral ring is the same in both compounds (**II**) and (**III**). However, quadrant rule can explain these effects. As shown in Figure 5c, where quadrant rule reveals that the methyl group at C-10 and greater part of the rest of the molecule lie in the upper left-hand sector, and therefore a negative first CD band is expected in the CD spectra (Table 1).

Interestingly, both 1st CD band (212 nm) and 2nd CD band (244 nm) of compound (**IV**) were appeared at lower wavelengths with positive and negative Cotton effects, respectively. Although, (**IV**) have no pseudoaxial substituents at the benzylic carbon atoms and have similar quinoxaline chromophore system, this shifting in CD band values could be attributed to the presence of electron withdrawing chloro substituent on quinoxaline chromophore. As there are three lone pairs on Cl, it can affect the transitions drastically, which can be explained by the appearance of both CD bands at lower CD bands wavelengths compared to (**II**) and (**III**). It will be interesting to study the effect of different substituents on CD as future research study. It will be interesting to study the effect of different substituents on CD as future research study. Also, to effectively deliver these steroidal drugs at specific target site, their conjugation with the biologically active nanomaterials will be an important strategy to consider [41–46].

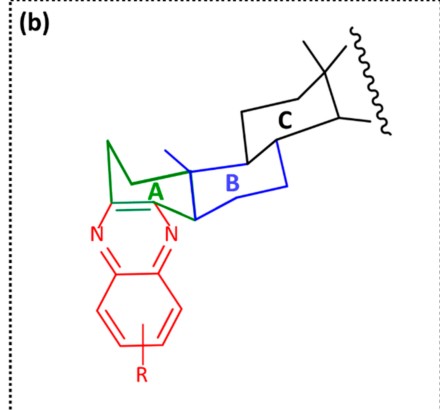

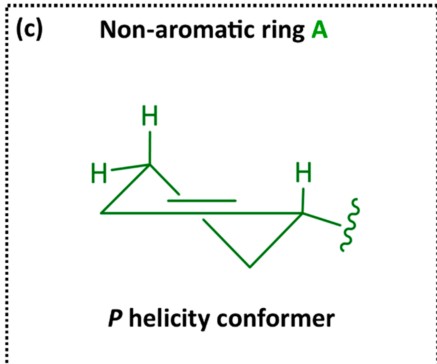

**Figure 5.** Chiral representation of steroidal quinoxalines: (**a**) Classification of steroidal quinoxa-lines into spheres for determining the helicity, (**b**) Stereo conformation of steroidal quinoxalines, (**c**) Quadrant rule applications to analyze CD data.

## 4. Conclusions

The present work demonstrates the structure-chirality relationship of synthesised steroidal quinoxalines, which are synthesised by condensing substituted and unsubstituted diaminobenzene at the α, β unsaturated ketone carbon positions at cholest-3-en-4-one under inert atmosphere reflux conditions. The final compounds were purified by column, thin layer, and preparative thin layer chromatography. Structural characterisation was carried out by different analytical, spectroscopy, spectrometry techniques including MP, FTIR, NMR, and MS. The CD spectra of unsubstituted and methyl-substituted quinoxalines have shown positive Cotton effects at 238 nm and 240 nm, respectively, which was used to determine their *P* helicity. Whereas the CD behaviour of chloro-substituted quinoxalines is quite different, and this may be attributed to presence of three lone pairs on Cl atom, which can affect the transitions drastically. It will be interesting to study the effect of different substituents on helicity of bio-active steroidal quinoxalines.

**Supplementary Materials:** The following are available online at https://www.mdpi.com/2624-854 9/3/1/30/s1. Full experimental detail, including 1H NMR, 13C NMR, MS, FTIR, and CD data.

**Author Contributions:** Methodology, investigation, writing, review, and editing, data curation visualization, formal analysis R.M.; Conceptualization, resources, formal analysis, and supervision N.R.; Formal analysis M.J.N.A., Formal analysis, S.U.; Formal analysis C.C.S. All authors have read and agreed to the published version of the manuscript.

**Funding:** This research received no external funding.

**Institutional Review Board Statement:** Not applicable.

**Informed Consent Statement:** Not applicable.

**Data Availability Statement:** Not applicable.

**Acknowledgments:** The authors wish to acknowledge Tibor Kurtán (Department of organic chemistry, university of Debrecen, Hungry) for providing analytical facilities. Authors also acknowledge the Higher Education Commission (HEC) Pakistan for research support and Hussain Ebrahim Jamal (HEJ) research institute of chemistry, university of Karachi Pakistan for the characterisation support.

**Conflicts of Interest:** The authors declare no conflict of interest.

**Sample Availability:** Samples of the compounds (**I**), (**II**), (**III**), and (**IV**) are available from the authors.

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
