# Peer review of "Synthesis and Structure-Chirality Relationship Analysis of Steroidal Quinoxalines to Design and Develop New Chiral Drugs"

_chemistry, doi:10.3390/chemistry3010030_

Round 1

Reviewer 1 Report

Peer-Review of Chemistry Manuscript chemistry-1066306

The manuscript entitled “Synthesis and Structure-Chirality Relationship Analysis of Steroidal Quinoxalines to Design and Develop New Chiral Drugs” describes the use of enantioselective chromatography and chiroptical measurements to study the racemization of the title reaction.

The study is mostly well presented, accurately supported by the included references and supporting information, and conclusions are in line with results obtained.

I believe that the work shown in the manuscript is of enough merit to be published in Chemistry. I only have a few minor comments that I think can improve the submitted manuscript before publication is granted.

  • The authors use empirical rules to establish the absolute configuration of the prepared compounds. Even though these rules appear to be used correctly, it is now widely established that such assignments should be confirmed using computational tools such as TD-DFT. In the case of the compounds presented, the size and conformational flexibility features that could hinder the use of such tools, should be easily overcome resorting to a truncated model that only have cycles A, B and C, as shown in Figure 5,b. Additionally, by using this approach, the authors should gain a better insight of the apparent odd behavior of compound IV.
  • Figure 3 doesn’t show the complete molecular structure of compounds II, III and IV.

Reviewer 2 Report

The authors report structure-chirality relationship of three new steroidal quinoxalines, which were synthesized by condensing diaminobenzenes with cholestenone. The structure was analyzed by circular dichroism (CD) technique. They concluded that unsubstituted and methyl-substituted quinoxalines, has P helicity.

The authors report some new compounds and they characterize the structure by CD spectra but correction and addition of some key information are necessary before the acceptance of the manuscript for publication.

Is the compound I truly a new compound? I think it is already reported in literature. The literature for the rest of the two was not found by SciFinder.

See J. Chem. Soc., 1956, 627-629 and J. Am. Chem. Soc. 1954, 76, 7, 1728–1733.

In line 15 and 192, “cotton effect” should be “Cotton effect”.

In line 150, it is written as follows. “Consequently, different alternative procedures were tried to overcome this problem. Of those procedures, the use of strong base, sodium hydride facilitated the condensation, resulting in good yield of the final products.” What are the alternative procedures?

In line 151, it is written as follows. “Of those procedures, the use of strong base, sodium hydride facilitated the condensation, resulting in good yield of the final products. ” The expression “good yield” is ambiguous, and it is better to provide specific value. I do not think 35 % yield of compound III is a good yield. Why the yield of compound III is low?

In Figure3, chemical structure of cholest-4-en-3-one is given. I think the stereochemistry of the compound is already known, so it is better to clarify the configuration of asymmetric carbon using broken wedge and solid wedge. In addition, waved line should be used to omit some part of the structure as has been done in Figure 2.

In the table, concentration of the compounds are given in unit of mol/L but the value seems to be too high. Apparently correct value is given in line 135-137.

In the table, the value of Cotton effect is given but the unit is not written.

In line 185, it is written as follows. “To determine the helicity, Snatzke and co-workers established that steroidal heterocycles can be divided into three spheres. According to them, if first and second sphere are achiral then sector rule or helicity rule can be applied.” Snatzke and co-workers uses one aromatic ring that is attached to the steroidal heterocycles and your sample uses quinoxaline. I think the transition electric dipole moments are different in each cases. Does the result of sector rule depend on the direction of the transition electric dipole moments?

In line 190, I cannot see some characters, possibly “pi”.

In line 196, it is written as follows. “Looking through quinoxaline chromophore, it can be seen in Fig 5(b) that ring A is in half chair conformation, whereas rings B and C maintain their chair conformation, resulting in molecular stability.” However, the configuration of each carbon is unclear and the conformation of ring A is not obvious. Author should consider modeling the compound using molecular mechanics or quantum mechanics calculation.

In line 198, you conclude ring A as II and III shows P-helicity which is predicted from 238 and 240 nm. I am not fully convinced that II and III shows P-helicity. Why the sign of the Cotton effect at 290 nm can be omitted? Sector rule is sensitive to the conformation of the compound and model need to be provided. Some calculation using DFT and CD spectra will be a good method to confirm your result.

Round 2

Reviewer 2 Report

The manuscript is improved by the authors but there are still some points that should be corrected.

I found the other typo “cotton effect” in Line 215.

In Figure3, the author clarified the configuration of asymmetric carbon at 10, 13 20-positions. However, carbons at 5, 8, 9, 14, and 17-positions are asymmetric carbon as well and their configuration should also be clarified because this information is mandatory to draw 3D image of the compound in Figure 5.

About the unit in the table, authors added the unit for wavelength but what is the unit for the Cotton effect? Is the unit in mdeg or M^(-1)cm^(-1)? Considering Figure 4, it seems in mdeg.
